# Stress-Related Chronic Fatigue Syndrome: A Case Report with a Positive Response to Alpha-Methyl-P-Tyrosine (AMPT) Treatment

**DOI:** 10.3390/ijms25147778

**Published:** 2024-07-16

**Authors:** Maria Ljungström, Elisa Oltra, Marta Pardo

**Affiliations:** 1Escuela de Doctorado, Universidad Católica de Valencia San Vicente Mártir, 46001 Valencia, Spain; maria.ljungstrom@mail.ucv.es; 2Department of Pathology, School of Health Sciences, Universidad Católica de Valencia San Vicente Mártir, 46001 Valencia, Spain; elisa.oltra@ucv.es; 3Department of Psychobiology, Universidad de Valencia, 46010 Valencia, Spain; 4Interuniversity Research Institute for Molecular Recognition and Technological Development (IDM), 46022 Valencia, Spain

**Keywords:** CFS (chronic fatigue syndrome), POTS (postural orthostatic tachycardia syndrome), stress, COMT (catechol-O-methyltransferase), adrenaline, inflammation, AMPT (alpha-methyl-p-tyrosine)

## Abstract

Chronic fatigue syndrome (CFS) is a heterogeneous disorder with a genetically associated vulnerability of the catecholamine metabolism (e.g., catechol O-methyltransferase polymorphisms), in which environmental factors have an important impact. Alpha-methyl-p-tyrosine (AMPT; also referred to as metyrosine) is an approved medication for the treatment of pheochromocytoma. As a tyrosine hydroxylase inhibitor, AMPT may be a potential candidate for the treatment of diseases involving catecholamine alterations. However, only small-scale clinical trials have tested AMPT repurposing in a few other illnesses. The current case report compiles genetic and longitudinal biochemical data for over a year of follow-up of a male patient sequentially diagnosed with sustained overstress, neurasthenia, CFS (diagnosed in 2012 as per the Center for Disease Control (CDC/Fukuda)), and postural orthostatic tachycardia syndrome (POTS) over a 10-year period and reports the patient’s symptom improvement in response to low–medium doses of AMPT. This case was recognized as a stress-related CFS case. Data are reported from medical records provided by the patient to allow a detailed response to treatment targeting the hyperadrenergic state presented by the patient. We highlight the lack of a positive response to classical approaches to treating CFS, reflecting the limitations of CFS diagnosis and available treatments to alleviate patients’ symptoms. The current pathomechanism hypothesis emphasizes monoamine alterations (hyperadrenergic state) in the DA/adrenergic system and a dysfunctional autonomic nervous system resulting from sympathetic overactivity. The response of the patient to AMPT treatment highlights the relevance of pacing with regard to stressful situations and increased activity. Importantly, the results do not indicate causality between AMPT and its action on the monoamine system, and future studies should evaluate the implications of other targets.

## 1. Introduction

Chronic fatigue syndrome (CFS) is a debilitating biological medical condition (International Classification of Diseases, ICD-11 8E49 [1]) with no approved treatment. It is characterized by unexplained physical fatigue, cognitive problems (brain fog), and post-exertional malaise (PEM) [2]. CFS’s etiology is controversial, and its underlying mechanisms are not properly understood [3]. Hypotheses of its pathophysiology include stress of the immune and neurobehavioral systems [4]. Dysregulation of the immune system and an altered hypothalamic–pituitary–adrenal (HPA) axis are recurrent findings in CFS [5]. As a heterogeneous disorder, the close focus and monitoring of an individual patient’s symptoms geared towards the identification of subtypes and clustering are needed for personalized treatments.

The most commonly referred cause of CFS is a viral infection, which is proposed to drive exhaustion of the immune system. Consistently, the literature supports a reduction in natural killer (NK) activity and anti-inflammatory cytokines, as well as increased levels of proinflammatory cytokines in CFS patients [6,7]. NK activity seems to indicate immune deficiency, with potential stratification value [8].

A subgroup of subjects diagnosed with CFS present stress-induced CFS [9,10]. Currently, there exists limited understanding regarding stress-related CFS. There is an association of early life exposure to stress in the development of CFS [11]. However, stressors during late adolescence or early adulthood (such as study/work-related stress) could also elicit CFS. The prevalence of stress-related CFS is unknown, and the exact onset mechanism remains unclear. However, the literature recognizes that stress can disrupt physiological systems and behaviors, contributing to the onset and perpetuation of CFS symptoms. The impact of perceived stress management on CFS symptoms has been studied, showing some benefits on physical symptoms such as PEM [12,13].

The HPA axis appears to be dysregulated in patients with CFS, thus explaining several symptoms of the disease, including PEM. A subset of CFS patients present hypocortisolism at awakening [14], but proper stress management seems to regulate cortisol levels and PEM in CFS to a certain extent [13]. Functional alterations of the glucocorticoid receptor gene (NR3C1) have also been connected to CFS [15].

The main catecholamines, dopamine (DA) and noradrenaline (NA), play an important role in HPA function and the response to stress. Some CFS cases and individuals with postural orthostatic tachycardia syndrome (POTS) often present with a hyperadrenergic state [16,17]. This “sustained” arousal may derive from a chronic response against infections as well as an adaptation to psychological stressors that interact with predisposing genetic or epigenetic factors [16]. Beta-adrenergic blockers are recommended for those cases [6,18]. However, some CFS patients with specific genetic alterations show a better response to beta-blocker treatment than others [6,19]. Main catecholamine regulators, such as the catechol-O-methyltransferase (COMT), have been shown to play an important role in CFS. A genetic variation in COMT has been recognized in a subset of CFS patients [19,20].

In addition to genetic factors, epigenetic modifications, such as differential DNA methylation at loci associated with differences in, for example, glucocorticoid sensitivity [5] or the COMT gene [21], may be important as biomarkers for future clinical testing of CFS.

The current case report follows the CARE guidelines for case reports [22]. We introduce a male patient (body weight 76–82 kg; height 185 cm) diagnosed with CFS in 2012 (Center for Disease Control (CDC/Fukuda) [2]), and recognized as a stress-related CFS case. Owing to the scarcity of studies specifically addressing this subgroup of CFS patients, the data acquired from the current patient are exceptionally valuable, contributing to an enhanced understanding of the underlying mechanisms involved in stress-related CFS. Throughout the patient’s medical records, multiple medical specialists conducted evaluations. They documented persistent and severe physical and mental fatigue lasting for over six months, recurrent headaches often described as mental fatigue, episodes of post-exertional malaise, disruptions or alterations in sleep patterns, and cognitive impairments. These cognitive issues were further detailed as a “cloudy mind,” encompassing difficulties with concentration and memory disturbances. The patient´s diagnosis later included POTS, with overlapping CFS symptoms. The genetics and biochemistry of the patient could indicate vulnerability and predisposition to stress-triggered disease, which, added to the personality traits of the patient, may explain a major lack of response to a considerable list of medications.

## 2. Case Description

### 2.1. Diagnostic Timeline

A 29-year-old male presented at the hospital in 2012, reporting severe fatigue and a history of exhaustion from 2008 to 2012, necessitating intermittent daytime naps. Upon admission, the patient self-reported symptoms of anxiety, stress, obsession, and narcissism, with a family history notable for psychiatric disorders (the mother diagnosed with bipolar disorder at age 40). The patient experienced stressors such as parental divorce, temporary separation from his mother, and work-related burnout. Social withdrawal, fear of criticism, and a tendency toward disappointment were also noted. Symptoms persisted for over six months, accompanied by pronounced physical and mental fatigue (napping 5–6 times daily), reduced concentration, diminished motivation, inability to work, and a depressed mood. Treatment initially focused on stress-management techniques, including psychotherapy, yoga, and meditation, which helped mitigate psychological stressors (2012). Elevated cortisol levels were observed during the diagnostic period (2012–2013), suggesting a hyperadrenergic state. In 2013, the patient received diagnoses of neurasthenia, exhaustion with narcissistic traits, depressive disorder, and subsequently CFS.

The patient reported gradual increases in fatigue and stress over the years. During the time frame 2021–2023, the patient provided MP with all personal and medical files and written informed consent for publication. Across the medical records, a range of symptoms indicative of dysfunction across multiple systems could be identified. We acknowledge the limited availability of data pertaining to the patient’s performance status at the time of diagnosis and throughout the disease course from 2013 to 2021. At that time, due to the patient’s symptoms associated with heightened arousal, he sought new medical professionals focusing on stress-related symptoms. In 2021, he attended a clinic specializing in chronic fatigue syndrome (CFS) and stress, where a genetic profiling study centered on the HPA axis and other stress-response genes relevant to CFS was conducted. From 2021 to 2023, the patient experienced post-exertional malaise (PEM) following moderate exercise, insomnia, reduced libido, heightened startle response, altered response to sweet taste, cognitive impairment (foggy brain), and mental fatigue. During 2021–2022, the patient experimented with nootropics, reversible inhibitors of monoamine oxidase A (RIMAs), L-tryptophan, and monoamine oxidase A inhibitors (Syrian rue). Additionally, he underwent an intensive supplement-based treatment regimen recommended by his physician. The patient described this period as challenging due to the discomfort of taking multiple supplements daily. While some medications initially alleviated symptoms for up to two weeks, tolerance quickly developed thereafter.

In 2023, the patient tried “Parasym”, a neuromodulation device for the vagal nerve shown to have therapeutic benefits [23]. The patient reported experiencing some improvement in their condition. In 2023, the patient was additionally diagnosed with POTS (Figure 1).

The patient underwent a structured clinical interview to provide clinical and socio-demographic treatment data from June 2022 to July 2023, while maintaining regular visits to their medical doctor. Most biochemical analyses related to the catecholamine system were conducted during this period. Biomolecule measurements at the end of 2021 (details in Table 1) may have been influenced by ongoing supplement use, which continued until early 2022. Despite treatment efforts, the patient’s health deteriorated, characterized by persistent physical fatigue. Zopiclone was prescribed from January to June 2023 to manage sleep, which stabilized average sleep duration at 7 h and reduced insomnia. The patient’s perceived quality of life declined notably, impacting social relationships significantly.

### 2.2. Biochemical and Genetic Analyses

Biochemical parameters included measuring circulating neurotransmitter levels. The potential involvement of the main catecholamines (DA and NA), as well as the role of glutamate and gamma-aminobutyric acid (GABA) neurotransmitters and related molecules were studied in blood/urine at different time points (see Table 1 for details and Appendix A for complete data).

In addition, a comprehensive panel of single-nucleotide polymorphisms (SNPs) (November 2021) (McMind company: https://shop.mcmind.de/, last accessed on 20 June 2024) showed the presence of some SNPs with potential relevance for the patient’s risk of developing CFS (Table 2; for complete data set, please see Appendix A). Some implied a direct impact on the regulation of the monoamines and neurotransmitters (COMT and MAO), and others indicated a direct involvement in the response to stress (e.g., NR3C1, 24) and methylenetetrahydrofolate reductase (MTHFR) [24] (Table 2). The change in wt “G” by an “A” in the SNP rs46080 of the COMT gene (Val158Met) is associated with lower COMT activity, which may lead to higher DA levels in the brain cortex, a reduced pain threshold, and an enhanced vulnerability to stress, which could align with the upper limit blood DA levels found in the patient (Table 1). Additional missense variants in heterozygosis: Ala222Val of the MTHFR leads to reduced processing of folic acid, and Val16Ala of the superoxide dismutase (SOD) gene translates into oxidative stress alterations. Importantly, some of these genetic conditions have previously been associated with CFS patients [25].

Further SNP analyses from Ancestry (https://www.ancestry.com/ last accessed on 20 June 2024) and 23andMe (https://www.23andme.com/gen101/snps/(2023) last accessed on 20 June 2024) were individually studied and classified for their role in cognitive processes, stress, motor disturbances (including physical fatigue), and inflammation, and the immune system (Appendix A).

In addition, the HPA axis and immune system composition and function were studied by measuring glucocorticoid and cytokine levels, as well as the presence of autoantibodies.

The patient presented with elevated proinflammatory cytokine levels (interleukin-6 (IL6), IL10, and interferon gamma (IFN_ϒ_) (2021–2022), indicative of neuroinflammation and a reduction in NK cells, suggesting a compromised immune system (Appendix A). In addition, the presence of Epstein–Barr virus (EBV) antibodies (Appendix A) and several receptor autoantibodies (Table 1) further supported immune problems. The patient´s immune system alterations are in agreement with the previously described CFS classical phenotype [6,7].

The itemized results for additional blood parameters, including blood counts, hormones, vitamins, etc., are broadly documented in Appendix A.

In summary, the patient presented altered parameters in neurotransmitter pathways and a hyperadrenergic state with increased levels of the main metabolites of serotonin and kynurenine, giving support to increased tryptophan catabolism. Increased levels of glutamate and GABA were confirmed, indicating aberrant system activity with altered autoregulation (Table 1).

## 3. Treatment and Course of Symptoms

After a decade of treatments with supplements and other approaches offering little or no help at all, drug treatments reported by the patient to MP (summer 2022–2023) seemed to have a basis to alleviate the patient’s hyperadrenergic state. Figure 2 summarizes the main treatments and adherence to those treatments. The lack of perceived symptom improvement significantly compromised the patient’s adherence to the prescribed treatment regimens.

Propranolol (a non-selective beta-adrenergic blocker) is currently an approved treatment for CFS [6] and has shown beneficial effects on CFS patients. The patient´s response to treatment was self-scored based on the patient’s daily notes (see below for more details). However, following increasing doses (20 mg bid/40 mg bid/60 mg bid), the patient reported increased rates of stress and fatigue across all doses (Figure 3A)).

When beta-blockers are not appropriate or irresponsive, moxonidine is recommended because it reduces the activity of the sympathetic nervous system via activation of I1-imidazoline receptors, resulting in a decrease in adrenaline and NA in humans [26]. With moxonidine, all symptoms worsened after a trial of increasing doses (0.2 mg bid/0.3 mg bid/0.4 mg bid). Additionally, a GABAb agonist (baclofen, 10 mg bid/20 mg bid/30 mg bid) did not improve mental and physical fatigue. Carvedilol (1.25 mg bid and 2.50 mg bid), an alpha-1 adrenergic receptor blocker, led to the same negative results. Methyldopa treatment (a centrally alpha-2 adrenergic agonist that inhibits the adrenergic neuronal outflow and depletes NA) led to extreme levels of stress and physical fatigue after only 2 days, making the patient also stop this treatment (1.25 mg bid). Detailed treatment information is provided in Figure 3.

Due to the patient’s prior inability to report his daily response to treatment in a more standardized and quantifiable way, MP created a simplified instrument to evaluate the patient´s response to the pharmaceutical treatments. What the patient described as his daily baseline levels of stress, physical fatigue, and mental fatigue were equal to 5. The data presented in Figure 3 and Figure 4 were standardized based on the brief daily notes of the patient, and his comparison to what he considered his basal level (represented as 5). The scale was categorized as follows: light/low/a bit/more = ±1, very/high = ±2, extreme = ±3, and very extreme = ±4, representing an improvement or worsening in his symptoms (differentiating between physical fatigue, mental fatigue, and stress, when provided).

However, following treatment failure to target his potential hyperadrenergic state, the patient reported to MP having experienced improvement with alpha-methyl-p-tyrosine or methyrosine (AMPT) at a dose of 80 to 300 mg bid, a compound that reduces the synthesis of NA and DA (Figure 4 and Figure 5).

Figure 4 shows the patient’s daily levels of stress, physical fatigue, and mental fatigue (compared to the patients’ baseline levels) across time in two separate periods of treatment with AMPT. As the patient felt better, he engaged in more physical activity that led to a few periods of PEM (as indicated on the graphs), which may explain increased levels of physical fatigue independent of treatment dose. Difficulty with compliance with treatment for personal reasons led to the interruption of AMPT treatment. However, symptom worsening drove the patient to restart medication a few weeks later. Although AMPT of doses 80–200 mg bid seemed efficient at reducing fatigue (trial 1), the patient reported self-increasing the dose in trial 2 in search of more drastic positive results. However, a higher dose (300 mg bid) did not improve the symptoms. The improvement of symptoms while taking AMPT led to post-exertional malaise (PEM), which appears to be a result of reported increased physical activity in response to feeling “better”. The patient did not report any additional side effects while taking AMPT. Doses of AMPT up to 1500 mg showed an absence of secondary effects [27], which supports the safety of the dose used by the patient (300 mg bid).

## 4. Materials and Methods

All data were provided by the patient.

### 4.1. Clinical Analysis

Clinical evaluation and diagnosis were performed by medical doctors.

Blood and urine samples were analyzed by several medical doctors across disease progression.

Genetic analyses of SNPs were performed by the following companies: McMind GmbH & Co (Bamberg, Germany), Ancestry (Dublin, Ireland), and 23andMe (Oss, The Netherlands).

Pharmacological treatments reported by the patient included: atenolol (Aliud Pharma, Laichingen, Germany), alpha-methyldopa (Stada, Barcelona, Spain), carvedilol (1A Pharma, Bayern, Germany), moxonidine (Ratiopharm, Ulm, Germany), propranolol (Aliud Pharma, Laichingen, Germany), mirtazapine (1A Pharma Bayern, Germany), zopiclone (Ratiopharm Ulm, Germany), and alpha-methyl-p-tyrosine (AMPT) (Angene International Limited, London, UK).

### 4.2. Self-Evaluation Assessment of Symptoms

Evaluation of pharmacological treatments: Due to the patient´s inability to report their daily response to treatment in a more standardized and quantifiable way, MP created a simplified instrument to evaluate the patient´s response to the pharmaceutical treatments. What the patient described as his daily baseline levels of stress, physical fatigue, and mental fatigue were equal to 5. The data presented in Figure 2 and Figure 3 were standardized based on the brief daily notes of the patient and his comparison to what he considered his basal level. The scale was categorized as follows: light/low/a bit/more = ±1, very/high = ±2, extreme = ±3, very extreme = ±4, representing an improvement or worsening of his symptoms (differentiating between physical fatigue, mental fatigue, and stress).

## 5. Summary and Conclusions

The current case report presents the case of a patient diagnosed with CFS over 10 years ago who was longitudinally followed for response to treatment. We highlight the lack of a positive response to classical approaches to treating CFS, reflecting the limitations of CFS diagnosis and available treatments to alleviate patients´ symptoms. Additionally, we show the lack of adherence of this CFS patient to medication (a common trend among CFS patients possibly, a consequence of improvement failure). The current case supports the implication of stress factors on the etiology of CFS as well as the role of catecholamines on the symptoms described.

The data obtained revealed biochemical indicators of an altered HPA axis as well as a genetic predisposition or vulnerability to stress, which, in other cases, could be of epigenetic nature. The current pathomechanism hypothesis highlights monoamine alterations (hyperadrenergic state) in the DA/adrenergic system and a dysfunctional autonomic nervous system resulting from sympathetic overactivity.

In the context of a hyperadrenergic state, dysfunctional autonomic nervous system (ANS) activity could arise primarily from excessive sympathetic overactivity. This condition would disrupt the delicate balance between the sympathetic and parasympathetic branches of the ANS, leading to a range of physiological disturbances. The patient presents excessive cathecolamine activity, which could implicate heightened sympathetic tone, which, in this case, is manifested clinically as altered temperature regulation and respiratory irregularities in response to limited physical exercise (reported by the patient). These symptoms reflect maladaptive responses in ANS regulation, potentially stemming from CNS dysregulation and abnormalities in neurotransmitter pathways such as DA and NE. The patient has been diagnosed with conditions associated with sympathetic overactivity-induced ANS dysfunction, such as POTS and stress-related disorders, which underscore the significant impact of hyperadrenergic states on overall autonomic function and health outcomes.

After years of a series of trial-and-error treatments, the patient highlights his unexpected positive response to AMPT. AMPT inhibits TH (converts tyrosine to DA) (Figure 5D). The patient had presented decreased levels of tyrosine and increased levels of DA and NA, as well as other downstream metabolites such as vanillylmandelic acid (VMA) and B1/B2 adrenergic autoantibody receptors (Table 1). Additionally, the tryptophan pathway showed elevated levels of quinolinic acid, a toxic product of activated microglia in the brain (Figure 5D). Due to the previous hyperadrenergic alteration history of the patient, it does not seem surprising that the patient showed reduced stress and fatigue levels after AMPT treatment (low–intermediate doses 80–300 mg bid). A limitation is not counting with post-AMPT treatment analytics.

AMPT treatment also attenuates oxidative stress and inflammation [28], aspects possibly altered in the patient according to his genotype. AMPT has been tested in clinical trials of 22q11.2 deletion syndrome (which is characteristic of COMT polymorphism) [27,29]. Other disorders have given AMPT a chance (obsessive-compulsive disorder [30], schizophrenia [31,32], and mood disorders [33], with some positive results). Currently, a metyrosine derivative is being tested for autism spectrum disorder (ClinicalTrials.gov Identifier: NCT05067582). We are not aware of previous studies that have reported the use of AMPT for the treatment of CFS patients. Our data support the relevance of personalized treatment, focused on the patient’s specific symptoms, biochemical findings (including circulating levels of relevant neurotransmitters), and genotype towards tailored improvement in the patient´s quality of life. CFS is a devastating heterogenic disorder with a lack of effective treatments.

The response to AMPT treatment highlights the relevance of pacing with regard to stressful situations and increased activity. We acknowledge some limitations of this study, as a case report reflects individual features and responses. We focus the current report on the discussion of data with a direct relationship to the main mechanisms related to AMPT treatment. It would have been informative to monitor the biochemical levels of the main catecholamines after AMPT treatment, which could have allowed us to establish the mechanistic causality of patient improvement. Current data are based on biochemical and genetic analyses and patient reports. Importantly, the results do not indicate causality between AMPT and its action on the monoamine system, and future studies should evaluate the implications of other targets. Additionally, we acknowledge that neurotransmitter levels in the blood are not the best indication of autonomic dysfunction itself.

This longitudinal study provides support for the hyperadrenergic hypothesis, possibly for a subset of patients, and provides, for the first time, positive results in response to AMPT. Our results open the possibility of AMPT repurposing through clinical trials suited for a specific subgroup of CFS patients with an altered DA/NA system and HPA axis function.

## Figures and Tables

**Figure 1 ijms-25-07778-f001:**
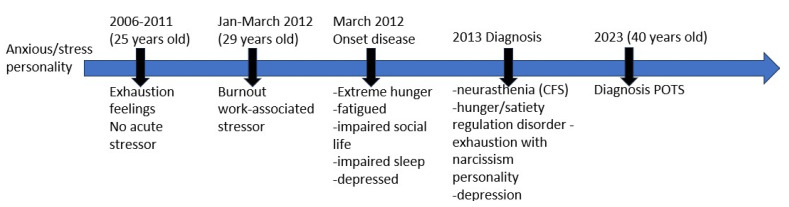
Timeline of medical history. The patient was under the care of multiple medical specialists from 2012 to 2023.

**Figure 2 ijms-25-07778-f002:**
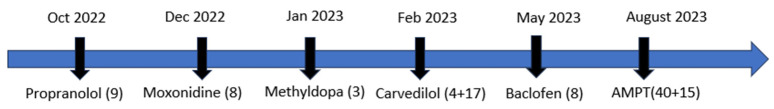
Timeline of pharmacological treatment. The patient was under several pharmacological treatments aimed at targeting the hyperadrenergic state from 2022 to 2023. Adherence (days) to each treatment is presented between parentheses.

**Figure 3 ijms-25-07778-f003:**
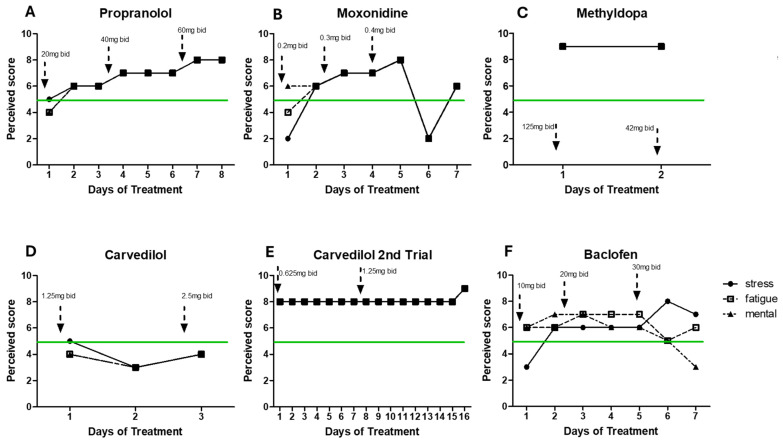
Response to various treatments. Depicts the time course of symptom levels (stress, physical fatigue, and mental fatigue). Dotted arrows indicate changes in dosage. The green line represents the patient’s baseline level. PEM = post-exertional malaise. (**A**) Propranolol, (**B**) moxonidine, (**C**) methyldopa, (**D**) carvedilol, (**E**) carvedilol 2nd trial, and (**F**) baclofen.

**Figure 4 ijms-25-07778-f004:**
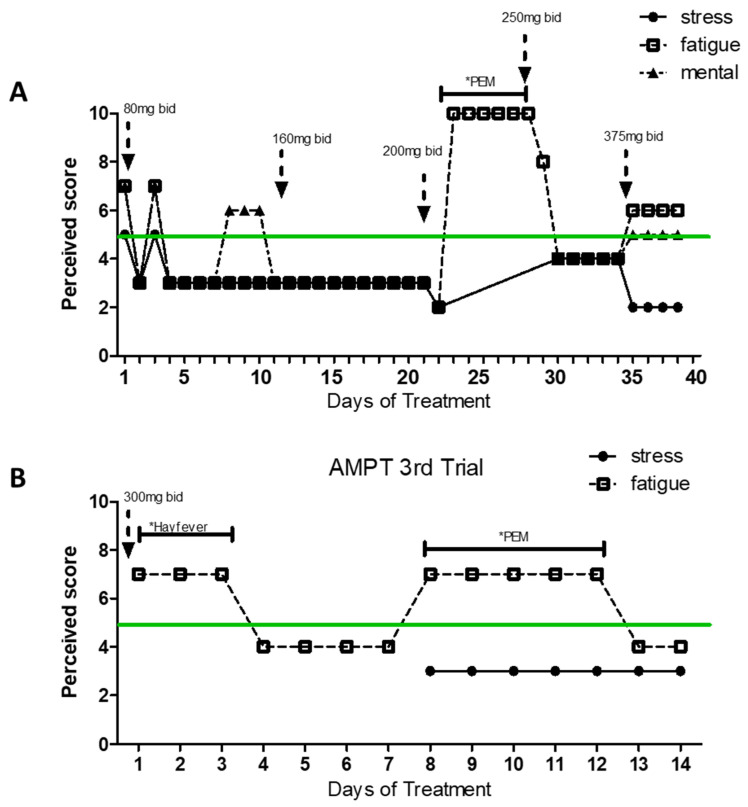
Response to AMPT treatment. Illustrates the time course of symptom levels (stress, physical fatigue, and mental fatigue) during AMPT treatment. Dotted arrows denote changes in dosage. The green line indicates the patient’s baseline level. PEM = post-exertional malaise. Panels (**A**) and (**B**) indicate separate trials/doses. * Specific condition as indicated appeared.

**Figure 5 ijms-25-07778-f005:**
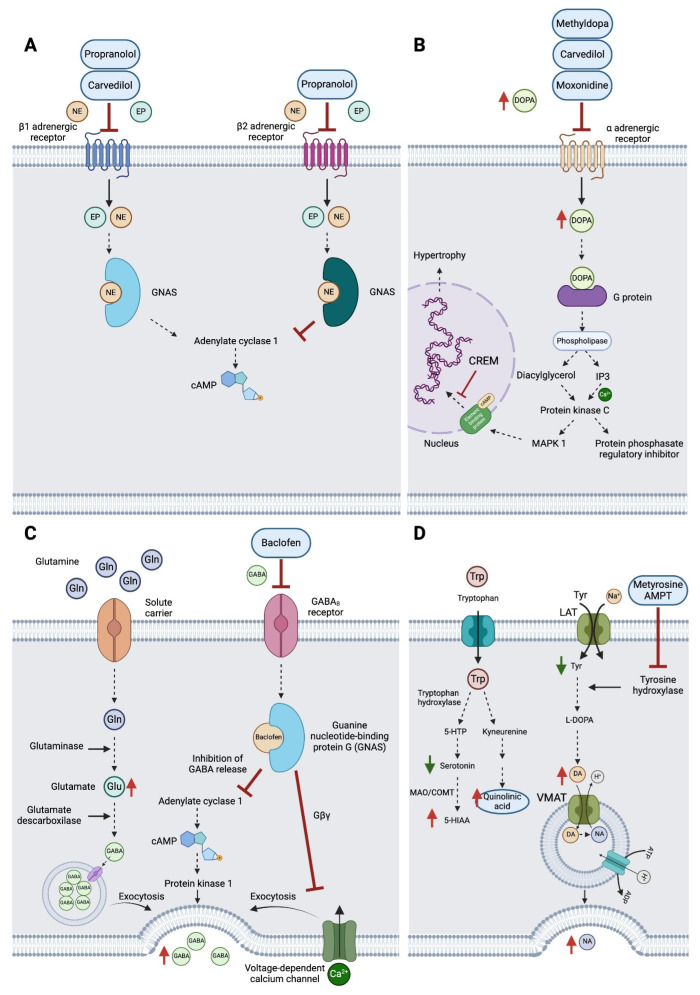
Molecular pathways affected and drug treatments. (**A**) Propranolol and carvedilol mechanism of action targeting β1 and β2 adrenergic receptors; (**B**) methyldopa, carvedilol, and moxonidine mechanism of action targeting α adrenergic receptors; (**C**) baclofen mechanism of action targeting GABA_B_ receptors; and (**D**) AMPT or metyrosine inhibition of tyrosine hydroxylase, affecting tyrosine metabolism and the related tryptophan pathway. Green (decrease) and red (increase) arrows represent patient´s data (created with Biorender.com, last accessed on 20 June 2024).

**Table 1 ijms-25-07778-t001:** Summary of main altered molecular parameters. The table displays the levels of biochemical parameters, their normal ranges, and the dates of measurement (neurotransmitters, autoantibodies, amino acids, and metabolites).

Biochemical Target	Normal Range	Unit	Level	Value	Date of Measurement
5-HIAA (5-hydroxyindoleacetic acid)	<2.40	mg/g	High	3.05 *	November 2021
Adrenaline	<20	μg	Normal	11	August 2012
			Normal	3.5	July 2022
			Normal	10.22	March 2023
B1 adrenergic receptor autoantibody	<15	U/mL	High	17.9 *	July 2021
B2 adrenergic receptor autoantibody	<8	U/mL	High	13.5 *	July 2021
Dopamine (plasma)	<85	ng/L	Normal	81	March 2023
GABA urine/creatinine	3–13.6	μmol/g	High	15.4 *	July 2022
Glutamate urine/creatinine	8–25	μmol/gCrea	High	42.4 *	July 2022
M3-muscarinic autoantibodies AChR	<6	U/mL	High	8.4 *	July 2021
Metanephrine	<350	μg	Normal	78	August 2012
			High	427 *	March 2023
Noradrenaline	<90	μg	Normal	77	August 2012
			Normal	19.5	July 2022
Noradrenaline	70–750 (lying)	ng/L	Normal	167	March 2023
Noradrenaline	200–1700 (standing)	ng/L	Normal	518	March 2023
Noradrenaline urine/creatinine	32–58	μg/gCrea	High	64.9 *	July 2022
Noradrenaline/adrenaline	3–6	quotient	Normal	5.6	July 2022
Normetanephrine	<600	μg	Normal	270	August 2012
Quinolinic acid	<5.50	mg/g	High	6.11 *	November 2021
Serotonin	50–20	μg/L	Normal	39.7	July 2022
Serotonin urine/creatinine	148–230	μg/gCrea	Low	131.9 *	July 2022
Taurine	5.4–31.3	mg/L	Low	4.8 *	July 2022
Tryptophan	7.30–12.50	mg/g	Normal	9.86	November 2021
			Normal	10.5	July 2022
Tyrosine	38.5–84.3	μmol/L	Low	35.18 *	November 2021
			Low	36.43 *	July 2022
VMA (vanillyl mandelic acid)	<3	mg/g	High	3.14 *	November 2021

(*) indicates values outside the normal range.

**Table 2 ijms-25-07778-t002:** Summary of main altered molecular parameters. Genetic single-nucleotide polymorphisms (SNPs) detected, gene names, SNP IDs, affected loci, and corresponding gene regions.

Gene Target	Locus (GRCh38)	rs ID	Genotype	Gene Region
COMT	22q11.21	rs4680	A/A homozygote	coding
MAOA	Xp11.3	VNTR30bp	High copy number	promoter
MAOB	Xp11.3	rs1799836	T/T homozygote	intron
FKBP5	6p21.31	rs1360780	C/T heterozygote	intron
FKBP5	6p21.31	rs9470080	C/T heterozygote	intron
FKBP5	6p21.31	rs4713916	G/A heterozygote	intron
FKBP5	6p21.31	rs9296158	G/A heterozygote	intron
NR3C1	5q31.3	rs6198	G/A heterozygote	3´UTR
MTHFR	1p36.22	rs1801133	C/T heterozygote	coding
SOD2	6q25.3	rs4880	C/T heterozygote	coding

Abbreviations: COMT (catechol-O-methyltransferase); MAOA (monoamine oxidase A); MAOB (monoamine oxidase B); FKBP5 (FKBP prolyl isomerase 5); NR3C1 (nuclear receptor subfamily 3 group C member 1); MTHFR (methylenetetrahydrofolate reductase); SOD2 (superoxide dismutase 2); GRCh38 (Genome Reference Consortium Human Build 38); rs (reference SNP cluster ID); VNTR (variable number of tandem repeat); UTR (untranslated region).

## Data Availability

All data are available in the article and additional Appendix A.

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
