# Peer review of "Stress-Related Chronic Fatigue Syndrome: A Case Report with a Positive Response to Alpha-Methyl-P-Tyrosine (AMPT) Treatment"

_ijms, 2024, doi:10.3390/ijms25147778_

Round 1

Reviewer 1 Report

Comments and Suggestions for Authors

[IJMS] Manuscript ID: ijms-3093102 - Review Request: “Case Report: Stress-related chronic fatigue syndrome: A case report with positive response to Alpha-methyl-p-tyrosine (AMPT) treatment”, authored by Ljungström, et al.

General Comments to the Authors:

This paper is a case report that details the responsiveness of various drugs and the efficacy of AMPT in a case of stress-related ME/CFS.  Given the limited number of case reports on ME/CFS, this case report would be a valuable contribution in this field.  However, there are some issues to be improved before publication:

Specific Comments:

1. It is believed that AMPT is not approved for use in ME/CFS.  It is essential to provide information on how ethical issues were resolved, as this will also be useful for the readers.

2. Please clearly state the diagnostic basis for ME/CFS and the diagnostic criteria applied.  Regarding the mental illnesses, such as depression diagnosed during the course, were they not excluded but treated as comorbidities?

3. The negative aspects of AMPT, including side effects and adverse events, should be discussed.  The authors should mention whether any adverse effects were observed in this patient.  Also, the dose setting of AMPT should be carefully disclosed.

4. There are some other mimicking agents like AMPT, such as doxazosin in pheochromocytoma treatments.  How were the effects using some alpha-blockade in this patient? 

5. Since the authors addressed the involvement of the HPA axis in the pathophysiology of ME/CFS.  The endocrine data including loading tests for the functional evaluation of HPA axis and sympathetic nervous systems should be assessed in this case before and after AMPT treatments.

Minor Comments:

1. If this is a case of ME/CFS, the performance status should be clarified at the time of diagnosis and during the course.

2. The introduction is somewhat verbose.  How about focusing on the relationship between ME/CFS, stress, and catecholamines?

3. Considering the number of drugs used during the course, how about including the drugs used in Figure 1?

4. What do the author think about the radiological approach using MIBG scintigraphy to know the catecholamine impact and the localization in such patient?

Author Response

Specific Comments:

  1. It is believed that AMPT is not approved for use in ME/CFS. It is essential to provide information on how ethical issues were resolved, as this will also be useful for the readers.

Response: We thank the reviewer for pointing this out. Currently, AMPT is approved for the treatment of pheochromocytoma, a health condition described to have hyperadrenergic state as that presented by the patient. In our article, we describe the current status of AMPT (line 18 and 362): “is an approved medication for the treatment of pheochromocytoma.”, “AMPT has been tested in clinical trials of 22q11.2 deletion syndrome (what is characteristic of COMT polymorphism) [28,30]. Other disorders have given AMPT a chance (obsessive compulsive disorder [31], schizophrenia [32,33] and mood disorders [34] with some positive results. Currently a metyrosine derivate is being tested for Autism Spectrum Disorder (ClinicalTrials.gov Identifier: NCT05067582)”.

Line 153 describes how the patient provided all personal and medical files to Dr. Pardo and response to different treatments. All data reported is based on the patient's reports and medical files provided to MP. We describe: line 153: “The patient underwent a structured clinical interview to provide clinical and socio-demographic treatment data from June 2022 to July 2023, while maintaining regular visits to their medical doctor”. The authors declare not medical knowledge to discuss ethical issues related to prescribed medications.

 We acknowledge that ethical issues would have been discussed and resolved with the specialized regular doctors that the patient visited. Based on line 223: “drug treatments reported by the patient to MP (summer 2022- 2023) seemed to have a basis to alleviate patient’s hyperadrenergic state”, and our data analyses and response to AMPT would give support to the use of AMPT to treat patient´s symptoms. After the patient´s reported improvement, line 268: “the patient reports to MP having experienced improvement with AMPT”, we presume that medical doctors got support for their hypothesis. It is important to highlight that this treatment plan was detailed for the individualized case presented. AMPT would not be recommended for CFS patients who would not have an hyperadrenergic state.

  1. Please clearly state the diagnostic basis for ME/CFS and the diagnostic criteria applied. Regarding the mental illnesses, such as depression diagnosed during the course, were they not excluded but treated as comorbidities?

Response: We have accordingly revised the medical files provided by the patient. Medical files do not give more detail on such diagnostic criteria. However, CDC defined CFS as meeting the following criteria:  the patient should have severe fatigue for more than six months as well as at least four of the following symptoms: a new type of headache or a change in the pattern or severity of the headache, myalgias, pain in multiple joints, post-exertional malaise lasting more than one day, sore throat, tender lymph nodes, unrefreshing sleep and significant impairment in short term memory or concentration. The modified CDC criterion was extensively used until 2015 (Sapra and Bhandari, 2023)

Sapra A, Bhandari P. Chronic Fatigue Syndrome. [Updated 2023 Jun 21]. In: StatPearls [Internet]. Treasure Island (FL): StatPearls Publishing; 2024 Jan-. Available from: https://www.ncbi.nlm.nih.gov/books/NBK557676/

Across medical files, the patient was evaluated by different medical specialists. They reported severe physical and mental fatigue for long periods of time (over six months), headaches (also reported as mental fatigue), experiences of post-exertional malaise, sleeping discomfort or alterations, and cognitive disabilities (also reported as cloudy mind, which included concentration and memory disturbances). Line 93 now includes this information as follows: “Throughout the patient's medical records, multiple medical specialists conducted evaluations. They documented persistent and severe physical and mental fatigue lasting for over six months, recurrent headaches often described as mental fatigue, episodes of post-exertional malaise, disruptions or alterations in sleep patterns, and cognitive impairments. These cognitive issues were further detailed as a "cloudy mind," encompassing difficulties with concentration and memory disturbances. ”.

Regarding comorbidities, no additional information was detected in the medical records. Treatment for depression is noted (in 2012 and 2013), and at the time of the CFS diagnosis in 2013, medical reports emphasized a psychological intervention for an eating disorder, which significantly alleviated the fatigue at that time. The patient reported not being under antidepressant treatment at the time of the consultation with Dr. Pardo.

  1. The negative aspects of AMPT, including side effects and adverse events, should be discussed. The authors should mention whether any adverse effects were observed in this patient. Also, the dose setting of AMPT should be carefully disclosed.

Response: We agree with the reviewer. Based on Dr. Pardo interviews with the patient, data was obtained to “quantify” patient's physical and mental fatigue as well as perceived stress (as reported in figures 2 and 3). As we discuss, line 250; “Due to the patient´s inability to report  daily response to treatment in a more standardized and quantifiable way, MP created a simplified instrument to evaluate patient´s response to the pharmaceutical treatments”. We acknowledge in line 275  “Difficulty for compliance with treatment for personal reasons led to interruption of AMPT treatment.” The patient did not report additional comments to Dr. Pardo. We verified doses reported by the patient, line 269: “dose of 80 to 300 mg bid”. Our review of the bibliography highlights that the doses reported falls on a safe range, line 283: “ Doses of AMPT up to 1500 mg showed absence of secondary effects [28], which supports the safety of the dose used by the patients (300 mg bid).” However, as previously detailed, Dr. Pardo has non medical training for dose indications. The patient reported visiting specialized medical doctors.

Based on reviewer's comment about side effects, we have added the following sentence on the text in regards of side effects, line 280: “The improvement of symptoms while taking AMPT, led to post-exertional malaise (PEM), which appears to be a result of reported increased physical activity in response to feeling "better." The patient did not report any additional side effects while taking AMPT”.

  1. There are some other mimicking agents like AMPT, such as doxazosin in pheochromocytoma treatments. How were the effects using some alpha-blockade in this patient?

Response: We thank the comment. The patient reported an initial trial with an alpha-blocker, carvedilol, prior to trying AMPT. Figure 3D shows response to Carvedilol. Based on patient´s comments to Dr. Pardo, we wrote, in line 244: “Carvedilol (1.25 mg bid and 2.50 mg bid), an alpha 1 adrenergic receptor blocker led to the same negative results. “ Patient reported only 3 days before dropping treatment due to discomfort and no improvement.. Line 239 to 249 reports all information provided about additional treatments targeting the noradrenergic system before AMPT.

  1. Since the authors addressed the involvement of the HPA axis in the pathophysiology of ME/CFS. The endocrine data including loading tests for the functional evaluation of HPA axis and sympathetic nervous systems should be assessed in this case before and after AMPT treatments.

Response: We agree with the reviewer. We discuss that limitation in our article.  line 359: “ A limitation being not counting with post- AMPT treatment analytics.” and line 377: “It would have been informative to monitor biochemical levels of the main catecholamines after AMPT treatment, what could have allowed establish mechanistic causality of patient improvement. ”

We provided as much data as available from all medical records. Supplementary table S1 and S2 have all detailed measures and dates. The patient ceased Dr. Pardo assistance after feeling better with AMPT and no further analyses have been performed to study the HPA axis or even the neurotransmitter levels since then (as reported by the patient in follow-up meetings). We strongly encourage the medical doctors to do such evaluation. For future studies, we acknowledge the need for additional data, in line 379: “Current data is based on biochemical and genetic analyses and patient´s reports. Importantly, the results do not indicate causality between AMPT and its action on the monoamine system and future studies should evaluate the implication of other targets.”

Minor Comments:

  1. If this is a case of ME/CFS, the performance status should be clarified at the time of diagnosis and during the course.

Response: We appreciate the comment. Medical records from the patient do not mention Myalgic Encephalomyelitis (ME). There is only mention to CFS, even though we acknowledge that ME/CFS and CFS are often used interchangeably in medical literature and by healthcare professionals. As previously specificated, Dr. Pardo is not the medical doctor that diagnosed and followed the course of the illness. Therefore, we only provided information available from medical records. We have revised the medical records and at the time of diagnosis the emphasis was made on the additional symptoms of the patient, including depression and eating disorder. Therefore, there are no additional details about the performance status. As indicated previously,  Line 93 now includes this information as follows: “Throughout the patient's medical records, multiple medical specialists conducted evaluations. They documented persistent and severe physical and mental fatigue lasting for over six months, recurrent headaches often described as mental fatigue, episodes of post-exertional malaise, disruptions or alterations in sleep patterns, and cognitive impairments. These cognitive issues were further detailed as a "cloudy mind," encompassing difficulties with concentration and memory disturbances. ”. Additionally, from 2013 to 2021, the patient did not report additional medical visits. He reports worsening of the symptoms and search of additional medical advice after 2021, as stated in line 130: “At that time, due to the patient's symptoms associated with heightened arousal, he sought new medical professionals focusing on stress-related symptoms. In 2021, he attended a clinic specializing in Chronic Fatigue Syndrome (CFS) and stress… “ Therefore, we have added the following sentence on the manuscript, line 128: “We acknowledge the limited availability of data pertaining to the patient's performance status at the time of diagnosis and throughout the disease course from 2013 to 2021. ”

  1. The introduction is somewhat verbose. How about focusing on the relationship between ME/CFS, stress, and catecholamines?

Response: Thank you for the suggestion. Diagnosing CFS is inherently complex due to its heterogeneous underlying mechanisms, which complicate the identification of effective treatments. Given the scarcity of cases specifically attributed to stress-related CFS, it is crucial to comprehensively report available data in the current case, focusing particularly on stress and the catecholamine system. The introduction has predominantly addressed stress and the catecholamine system (lines 59-85), with only a brief mention (lines 54-58) of characteristic features observed in viral-related CFS cases, such as impaired NK cell levels and inflammation. It is pertinent to elucidate shared aspects between this case and viral-related CFS, as both exhibit overlapping mechanistic disruptions in biochemistry. For instance, the patient manifests altered NK cell levels and inflammatory markers, underscoring the relevance of exploring these similarities in understanding the pathophysiology and potential therapeutic strategies of CFS.

  1. Considering the number of drugs used during the course, how about including the drugs used in Figure 1?

Response: Thank you for the suggestion. Figure 1 attempts to summarize the timeline of the patient's medical history. We believe that incorporating all drug treatments into this figure could obscure its intended purpose. However, following reviewer´s request, a new figure 2 detailing the specific timeline of treatments has been added in section 3, starting in line 224.

 “Figure 2 summarizes main treatments and adherence to those treatments. The lack of perceived symptom improvement significantly compromised the patient's adherence to the prescribed treatment regimens.

Figure 2. Timeline of Pharmacological treatment. The patient was under several pharmacological treatments aimed to target the hyperadrenergic state from 2022 to 2023. Adherence (days) to each treatment is presented between parenthesis.”

  1. What do the author think about the radiological approach using MIBG scintigraphy to know the catecholamine impact and the localization in such patient?

Response: We really appreciate the reviewer comment. MIBG scintigraphy can be valuable in evaluating the CFS patient with the described hyperadrenergic state by providing detailed assessment of sympathetic nervous system activity and aiding in the exclusion of underlying catecholamine-secreting tumors such as pheochromocytomas or paragangliomas. This imaging modality facilitates targeted therapeutic decision-making and monitoring of treatment efficacy by quantifying adrenergic dysfunction. Additionally, MIBG scintigraphy could contribute to the understanding of CFS pathophysiology of the case, particularly in relation to autonomic dysregulation, and aids in phenotyping the patient for more tailored management approaches. Additionally, we acknowledge the need to screen for tumors on the patient and this technique could provide really useful information.

Reviewer 2 Report

Comments and Suggestions for Authors

Dear Authors,

the manuscript I have been glad to review presents an interesting and relevant clinical case especially considering that chronic fatigue syndrome is a surprisingly common and extremely debilitating disease, which treatment is often unsuccessful in restoring the patient’s quality of life.

Although the manuscript is well written and well structured, I suggest a few minor changes to be made. The materials and methods section should be divided in paragraphs separating the clinical analysis from the self-evaluation of the symptoms through the newly designed method (which is properly described). Moreover, I think more information (if available) should be given regarding the institutions or companies where the blood and urine samples were analyzed. Similarly, since the companies involved in the genetic testing offer a various array of products and tests, more details regarding the genetic testing would significantly improve transparency and readability of the manuscript. As my last observation, a little editing is needed on the captions of figures and tables and their spacing with the text.

Ultimately, I have suggested this manuscript to be published after minor revision because of its novelty and value in the current literature about chronic fatigue syndrome. 

Author Response

Although the manuscript is well written and well structured, I suggest a few minor changes to be made.

  1. The materials and methods section should be divided in paragraphs separating the clinical analysis from the self-evaluation of the symptoms through the newly designed method (which is properly described).

Response: We followed reviewer´s suggestion and have divided the materials and methods into separated subsections as follows, in line 302 and 313: 4.1 Clinical analysis, 4.2 Self-evaluation assessment of symptoms.

  1. I think more information (if available) should be given regarding the institutions or companies where the blood and urine samples were analyzed. Similarly, since the companies involved in the genetic testing offer a various array of products and tests, more details regarding the genetic testing would significantly improve transparency and readability of the manuscript.

Response: We thank the comment. The patient requested complete anonymity to preserve the privacy of both the healthcare providers and themselves. In adherence to this request, we included comprehensive details from all accessible medical records to ensure transparency and accuracy of the data presented. Currently, we prefer to maintain the anonymity of the patient and the involved entities. It is important to note that the data presented in this study has been objectively collected and is not selectively biased towards the objectives of the paper. To enhance transparency, we included two supplementary tables containing all available data, including specific details such as the dates of data collection and the names of the companies that provided the genetic results.

  1. A little editing is needed on the captions of figures and tables and their spacing with the text.

Response: We appreciate the comment. Following reviewer observation, we revised the captions and we modified them as follows:

-Figure 1, Line 150: “Timeline of Medical History. The patient was under the care of multiple medical specialists from 2012 to 2023”

-Table 1: line 163 now states: “Table 1. Summary of Main Altered Molecular Parameters. Displays levels of biochemical parameters, their normal ranges, and the dates of measurement (neurotransmitters, autoantibodies, amino acids, and metabolites).

(*) indicates values outside the normal range."

-Table 2: line 189 now says: “Table 2. Summary of Main Altered Molecular Parameters. Genetic single nucleotide polymorphisms (SNPs) detected. Displays gene names, SNP IDs, affected loci, and corresponding gene regions.”

-Figure 3: line 261 now states: “Response to Various Treatments. Depicts the time course of symptom levels (stress, physical and mental fatigue). Dotted arrows indicate changes in dosage. The green line represents the patient's baseline level”.

-Figure 4, line 287 has been rephrased: “Illustrates the time course of symptom levels (stress, physical, and mental fatigue) during AMPT treatment.  Dotted arrows denotes changes in dosage. The green… “

4.Ultimately, I have suggested this manuscript to be published after minor revision because of its novelty and value in the current literature about chronic fatigue syndrome.

Response: We really appreciate reviewer input

Reviewer 3 Report

Comments and Suggestions for Authors

Author Response

The authors should consider the following comments:

Abstract a. The abstract should contain the study objective, a brief description of the methods, and the conclusion. The authors should provide more details about the case study.

Response: We appreciate reviewer´s comment. Following reviewer´s request, we have rephrased the abstract as follows: line 16: “Chronic fatigue syndrome (CFS) is an heterogenous disorder with genetically associated vulnerability of catecholamine metabolism (e.g. catechol O-methyltransferase polymorphisms) holding an important impact of environmental factors. Alpha-methyl-p-tyrosine (AMPT; also referred as metyrosine) is an approved medication for the treatment of pheochromocytoma. As a tyrosine hydroxylase inhibitor AMPT may be a potential candidate for the treatment of diseases involving catecholamine alterations. However, only small-scale clinical trials have tested AMPT repurposing in few other illnesses. The current case report compiles genetic and longitudinal biochemical data for over a year follow-up of a male patient sequentially diagnosed with sustained overstress, neurasthenia, CFS (diagnosed in 2012 (Center for Disease Control (CDC/Fukuda)), and POTS (postural orthostatic tachycardia syndrome) over a 10-year period, and reports patient’s symptom improvement in response to low-medium doses of AMPT. This case was recognized as a stress-related CFS case. Data is reported from medical records provided by the patient to allow detailed response to treatment targeting the hyperadrenergic state presented by the patient. We highlight the lack of positive response to classical approaches to treat CFS, reflecting the limitations of CFS diagnosis and available treatments to alleviate patients´ symptoms. The current pathomechanism hypothesis emphasizes monoamine alterations (hyperadrenergic state) on the DA/adrenergic system and a dysfunctional autonomic nervous system resulting from sympathetic overactivity. The response to AMPT treatment on the patient highlights the relevance of pacing regarding to stressful situations and increased activity. Importantly, the results do not indicate causality between AMPT and its action on the monoamine system and future studies should evaluate the implication of other targets.”.

 Introduction a. The introduction provides general background information regarding factors, including catecholamine regulators, that have been shown to play an important role in Chronic Fatigue Syndrome (CFS). b.

 The authors should better describe the gap in the literature, the novelty, and the significance of the study. The language usage is easy to read.

Response: We followed the reviewer´s request and have included several statements across the introduction to highlight the gap, the novelty and the significance. We have added the following sentences: line 60: “Currently, there exists limited understanding regarding stress-related CFS.”,  and line 90: “Owing to the scarcity of studies specifically addressing this subgroup of CFS patients, the data acquired from the current patient are exceptionally valuable, contributing to an enhanced understanding of the underlying mechanisms involved in stress-related CFS. “

Methods a. In the case description, the full names of the psychologists are not necessary. The authors should limit sentences that contain quotes.

Response: We appreciate reviewer´s comment. The main author of the manuscript, Dr. Pardo, obtained the data directly from the patient. The patient reached out to Dr. Pardo due to her expertise on the catecholamine system and the stress response. We mentioned the psychologist only because we think it is important for transparency to show that data was directly provided to the authors (and psychologist at the same time), data was provided by the patient during his interviews with Dr. Pardo.

The patient requested complete anonymity to preserve the privacy of both the healthcare providers and themselves. In adherence to this request, we included comprehensive details from all accessible medical records to ensure transparency and accuracy of the data presented. It is important to note that the data presented in this study has been objectively collected and is not selectively biased towards the objectives of the paper. To enhance transparency, we included two supplementary tables containing all available data, including specific details such as the dates of data collection and the names of the companies that provided the genetic results.

Following reviewer´s comment, we have rephrased the text to avoid mentioning the psychologist, as follows: line 125 ”During the time frame 2021-2023 the patient provided MP all personal and medical files and written informed consent for publication”. We now avoid any mention of a psychologist's name.

The language of the manuscript (case description) should be revised to be more neutral and scientific. b. It is recommended to add the diagnostic criteria for CFS.

Response: The language of the case description has been revised to be more scientific, and all case description section (line 106-161) has been slightly rephrased to be more neutral and scientific as requested.

 Line 93 now introduces criteria fulfilled for its diagnosis with CFS (CDC/Fukuda).: “Throughout the patient's medical records, multiple medical specialists conducted evaluations. They documented persistent and severe physical and mental fatigue lasting for over six months, recurrent headaches often described as mental fatigue, episodes of post-exertional malaise, disruptions or alterations in sleep patterns, and cognitive impairments. These cognitive issues were further detailed as a "cloudy mind," encompassing difficulties with concentration and memory disturbances.”

Results a. The tables and Figures are clear. The results present the minimum necessary information about the tested parameters.

Response: We thank the reviewer´s observation.

Discussion a. The results have been discussed in the context of previous studies. The authors frequently refer to the term "hyperadrenergic state." Please explain the dysfunctional autonomic nervous system resulting from sympathetic overactivity in the discussion

Response: We thank reviewer´s comment. We now have added the following detailed explanation of the autonomic nervous system (ANS) dysfunction related to the hyperadrenergic state. In line 338: “In the context of a hyperadrenergic state, dysfunctional autonomic nervous system (ANS)  activity could arise primarily from excessive sympathetic overactivity. This condition would be disrupting the delicate balance between the sympathetic and parasympathetic branches of the ANS, leading to a range of physiological disturbances. The patient present excessive cathecolamine activity which could implicate heightened sympathetic tone, that, in this case is manifested clinically as altered temperature regulation, and respiratory irregularities in response to limited physical exercise (reported by patient). These symptoms reflect maladaptive responses in ANS regulation, potentially stemming from CNS dysregulation and abnormalities in neurotransmitter pathways such as DA and NE. The patient has been diagnosed by conditions associated with sympathetic overactivity-induced ANS dysfunction, such as POTS and stress-related disorders, which underscore the significant impact of hyperadrenergic states on overall autonomic function and health outcomes.”

Round 2

Reviewer 1 Report

Comments and Suggestions for Authors

The authors have  responded all previous comments and concerns thoroughly.  I have no further comments or suggestions.

Reviewer 3 Report

Comments and Suggestions for Authors

I accept in present form.The authors have revised the manuscript according to the given recommendations.I have no additional comments.